

# Diffraction of strongly interacting molecular Bose-Einstein condensate from standing wave light pulses

**Qi Liang, Chen Li, Sebastian Erne, Pradyumna Paranjape,
RuGway Wu and Jörg Schmiedmayer⋆**

Vienna Center for Quantum Science and Technology (VCQ),
Atominstitut, TU Wien, Vienna, Austria

⋆ Schmiedmayer@atomchip.org

## Abstract

We study the effects of strong inter-particle interaction on diffraction of a Bose-Einstein condensate of $^6Li_2$ molecules from a periodic potential created by pulses of a far detuned optical standing wave. For short pulses we observe the standard Kapitza-Dirac diffraction, with the contrast of the diffraction pattern strongly reduced for very large interactions due to interaction-dependent loss processes. For longer pulses diffraction shows the characteristic for matter waves impinging on an array of tubes and coherent channeling transport. We observe a slowing down of the time evolution governing the population of the momentum modes caused by the strong atom interaction. A simple physical explanation of that slowing down is the phase shift caused by the self-interaction of the forming matter wave patterns inside the standing light wave. Simple 1D mean field simulations qualitatively capture the phenomenon, however to quantitatively reproduce the experimental results the molecular scattering length has to be multiplied by factor of 4.2. In addition, two contributions to interaction-dependent degradation of the coherent diffraction patterns were identified: (i) in-trap loss of molecules during the lattice pulse, which involves dissociation of Feshbach molecules into free atoms, as confirmed by radio-frequency spectroscopy and (ii) collisions between different momentum modes during separation. This was confirmed by interferometrically recombining the diffracted momenta into the zero-momentum peak, which consequently removed the scattering background.

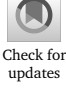

# 1 Introduction

Matter-wave diffracting from a standing wave of light demonstrates the fundamental concept of wave-particle duality. As first predicted by Kaptiza and Dirac [1], when particles move through a standing wave of light, they undergo two-photon scattering processes and gain discrete momenta in units of the vector sum of the two photons' recoil. On the other side one can see the diffraction as coming from the phase imprinted on the matter wave by the dipole potential of the standing wave. For a detailed discussion see: [2]. The first experimental observation of this phenomenon was made with thermal atom beams [3–5] and later with electron beams [6,7] and Bose-Einstein condensates (BEC) [8–11]. Moreover diffraction from a standing wave and the associated two photon transitions are an essential building block of atom interferometry and have enabled numerous fundamental tests, precision measurement and opened up many practical applications like inertial sensors [12,13], gravimeters [14–16], measuring the gravitational constant [17, 18], and were proposed to be used for detecting gravitational waves [19, 20].

The effect of inter-particle interaction during the diffraction process was in most cases neglected, mainly for its minor significance during the short time of the diffraction process and to obtain mathematical simple results. However such simplification may no longer guarantee accurate results for strong interaction or long timescales [21–23]. In this paper, we report an experimental study on diffraction of a molecular Bose-Einstein condensate (mBEC) of $^6Li_2$ from a standing light wave in the presence of strong inter-particle interaction. Due to the fermionic nature of $^6Li$ atoms, inelastic processes are strongly suppressed in a $^6Li_2$ molecular BEC [24,25], allowing for the preparation of initial equilibrium many-body states with strong interaction for the experiments. By tuning the $s$-wave scattering length with a magnetic Feshbach resonance, we are able to study the influence of interaction on the diffraction process over a wide range. Numerical simulations were performed to provide comparisons with experimental observations, and to assist in testing and understanding the physical processes.

# 2 Experimental procedures

The experiments are performed with BECs of $^6Li$ Feshbach molecules. Our experimental setup and procedure are detailed in Appendix A.1. In brief, in each experimental cycle we pre-

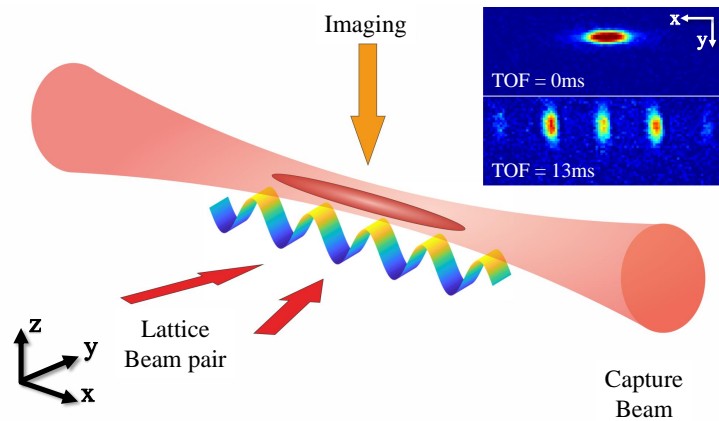

Figure 1: Schematic of the experimental setup with images taken before and after focusing.

pare a two-state mixture of lithium atoms in the lowest hyperfine states $|F = 1/2, m_F = 1/2\rangle$ (state $|1\rangle$), $|F = 1/2, m_F = -1/2\rangle$ (state $|2\rangle$), which correspond to $|m_I = 1, m_s = -1/2\rangle$ and $|m_I = 0, m_s = -1/2\rangle$ at high magnetic field, where the experiments are performed..Then by evaporative cooling in a single beam dipole trap on the BEC side (780 G) of the 832 G Feshbach resonance [26] the atoms form weakly bound Feshbach molecules each consisting of two atoms in different hyperfine states. With further evaporation, the molecules subsequently form a mBEC.

The procedure prepares mBECs of $\sim$3000 $^6\text{Li}_2$ Feshbach molecules [27]. The $s$-wave scattering length between molecules is tuned by setting the magnetic field [28]. For weakly bound molecules close to the Feshbach resonance, the dimer-dimer $s$-wave scattering length is given by $a_{dd} = 0.6a_{12}$ [24], where $a_{12}$ is the scattering length between atoms in states $|1\rangle$ and $|2\rangle$.

Figure 1 shows a qualitative sketch of the setup. The mBEC is confined in a trapping potential formed by a focused laser beam (the capture beam) and the magnetic field curvature produced by the electric coils. The combined potential provides trap frequencies $(f_x, f_y, f_z) = (16, 74, 68)\,\text{Hz}$, where $x$ denotes the axial direction along the trapping beam, $y$ the other horizontal direction, and $z$ the vertical direction. The field curvature is confining horizontally and therefore enhances trapping along the axial direction. Over the range of magnetic fields used, the trap frequencies are only very weakly affected by changing the field level. The axial trap frequency is varied by 7%, for magnetic field offset from 650 G to 750 G. The radial directions are dominated by the optical dipole trap.

To perform diffraction, a lattice potential $U(x) = U_0 \cos^2(\pi x/D)$ is formed with two crossing laser beams, where $U_0$ is the lattice potential depth, $D = \lambda/2\sin(\theta/2)$ the spatial period with wavelength $\lambda = 1064\,\text{nm}$ and the laser beam crossing angle $\theta = 15°$, resulting in a lattice period $D = 4\,\mu\text{m}$. The lattice laser beams are focused to beam waists of $w_{horizontal} = 600\,\mu m$ and $w_{vertical} = 140\,\mu m$. The beams are large compared to the size of the mBEC, and hence the lattice potential depth is approximately uniform across the cloud. The two lattice laser beams are derived from the same laser source, intensity controlled by an acousto-optic modulator (AOM), and subsequently split by a 50:50 beam splitter. The recoil energy of single photon transition is $E_r = \hbar^2 k^2/2m \approx 250\,\text{Hz}$, where $k = \pi/D$ and $m$ is the mass of a lithium molecule ($^6\text{Li}_2$).

The lattice potential is pulsed on for a variable time $t$ while the mBEC is held in the trap, such that a well-defined geometry and interaction energy is maintained during the scattering process. Immediately afterwards the cloud is released and allowed to expand. The cloud expands rapidly along the radial directions, hence quickly reducing interaction, while the mag-

netic coils are kept on such that the field curvature provides a focusing potential in the horizontal directions during the time-of-flight to measure the momentum distribution [29]. After a quarter of the oscillation period set by the horizontal trapping frequency of 16 Hz, the initial position distribution has collapsed, and the spatial pattern corresponds to the momentum distribution in the $x$ and $y$ directions before trap release. Detection is then performed by absorption imaging.

# 3 Experimental observations

## 3.1 Within Raman-Nath regime

For a pulse time $t$ much shorter than the oscillation period of a molecule in the lattice site, or equivalently $t\sqrt{E_r U_0}/\hbar < 1$, the particles remain approximately stationary during the lattice pulse. This regime is referred to as thin grating approach, also known as the Raman-Nath approximation [30]. The lattice potential could be viewed as a spatially periodic phase imprinting on the condensate wavefunction, resulting in the interference pattern in the far field, with bright fringes corresponding to momentum modes with a spacing of $2\hbar k$. The probability of finding atoms in the $n^{th}$ diffracted state is given by the Fourier transform of the imprinted phase shift. The occupation of the momentum modes ($\pm 2n\hbar k$) is hence given by Bessel functions of the first kind: $P_{\pm n} = J_n^2(tU_0/2\hbar)$ [31].

The lattice is pulsed with $U_0 = 500E_r$ and a duration of $t = 0 \sim 20\,\mu s$. Within this time range, the condition for Raman-Nath regime is satisfied, $t_{max}\sqrt{E_r U_0}/\hbar \sim 0.7$. As shown in Figure 2, distinct momentum modes can be recognized from the absorption images, while a strong background appears for high interaction strengths. The enhanced presence of the broad background with increased interaction suggests an interaction dependent loss of molecules from the condensate.

In order to determine the populations in each momentum mode, we integrate the images over the $y$ direction and fit a dual-component function

$$\sum_i A_i^c e^{\left(\frac{x-\mu+id_{sep}}{\sigma^c}\right)^2} + \sum_j A_j^g e^{\left(\frac{x-\mu+2jd_{sep}}{\sigma^g}\right)^2}.$$

Here $A$ and $\sigma$ denote the amplitude and width of the corresponding peak, where the superscript $c$ denotes the molecular condensate and $g$ the incoherent background. $d_{sep}$ is the spatial separation corresponding to the momentum $2\hbar k$ which can be accurately determined. Thus

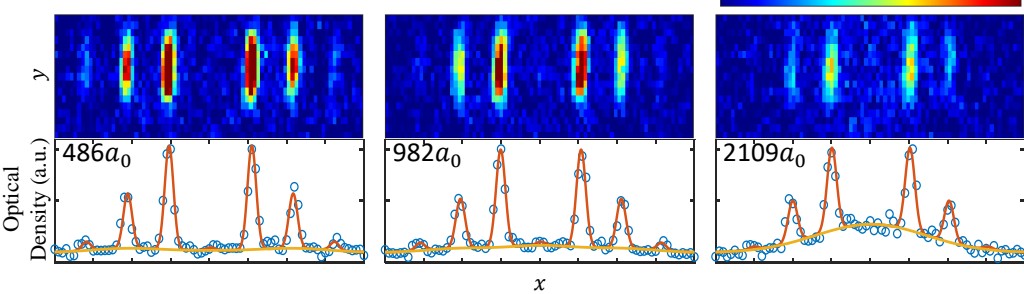

Figure 2: *(top)* Images taken with pulse $U_0 = 500E_r$ and $t = 5\mu s$ at different interaction strength and *(bottom)* the corresponding fitting result. Circles show the experiment data. Red and yellow solid lines shows the fitting result of the momentum peaks and background respectively.

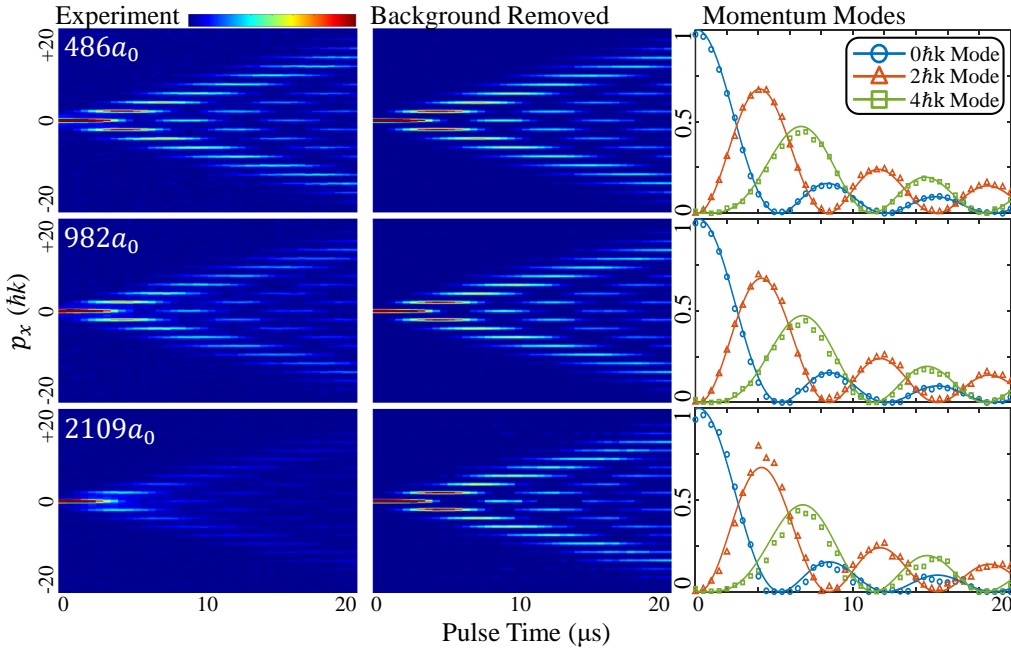

Figure 3: *(left)* Time carpets measured with $U_0 = 500E_r$ at different interaction strengths. *(middle)* the normalized momentum modes with background removed. *(right)* Normalized populations of the first three momentum modes (open symbols) for different dimer-dimer scattering lengths $a_{dd}$ are plotted, together with the corresponding Bessel functions calculated with the calibrated lattice intensity with no free parameters (solid lines). For the carpet plots we adjusted the color bar range to optimize the visual contrast. The carpet plot value corresponds to the optical density, in arbitrary unit.

we can determine the population of condensate peaks and scattered background separately. The integrated profiles showing the momentum mode occupations are plotted in a time carpet over the pulse time to provide an overview of the diffraction process in momentum space (Figure 3(left)). For further analysis, we remove the background from the dual-component fitting, time carpets can then be presented clearly with the momentum mode populations normalized to the total condensed population (Figure 3(middle)).

The time evolution of the $0\hbar k$, $\pm 2\hbar k$, and $\pm 4\hbar k$ populations are shown together with the theoretical results given by the Bessel functions (Figure 3(right)). For stronger interactions, despite significant losses, the normalized populations are found to still agree quite well with theory.

## 3.2 Beyond Raman-Nath regime

For longer pulse durations, the displacement of particles during the lattice pulse becomes non-negligible, thus the stationary approximation is no longer valid. Beyond the Raman-Nath regime, if the lattice depth satisfies the condition $U_0 \gg E_r$, the situation is referred to as the channeling regime [32]. The particles oscillate within each lattice site, creating a periodic pattern in momentum space. An analytical solution is obtained for the weak-pulse limit [33], but in general the population evolution during the scattering process needs to be calculated numerically.

Figure 4 shows the results measured with $U_0 = 50E_r$, and $t = 0 \sim 1000\,\mu$s. The weaker lattice restricts the particles within the first five momentum modes to achieve a decent imaging

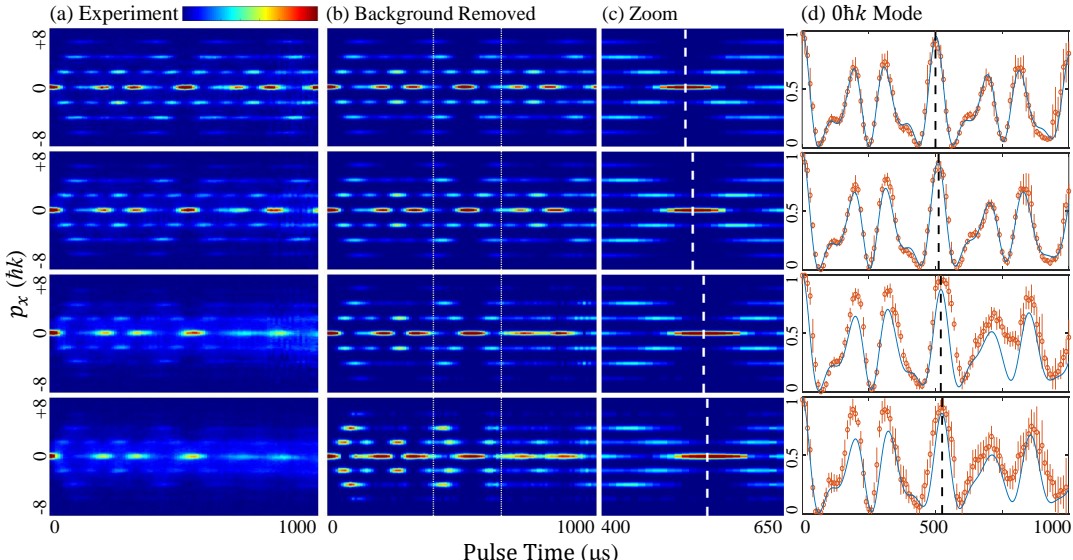

Figure 4: *(a)* Time carpets measured with $U_0 = 50E_r$ for different interaction strengths. *(b)* The momentum modes with background removed and normalized to the total condensate number at each time. The vertical dash lines mark the zoom in range for *(c)*. *(d)* The corresponding normalized populations of the $0\ \hbar k$ momentum mode (circle), shown in each case with the 1D GPE simulation result (solid line). The error bars for the population show the standard deviation of the fit results from 10 measurements. The simulation curves were produced with mean field interaction, including a phenomenological scaling factor $\eta = 4.2$ common to all case to take into account the additional interaction energy, as explained in the text. The vertical dashed lines in *(c)* and *(d)* indicate the time point at which the coherent population is restored to the $0\hbar k$ mode (recurrence of the mBEC). The shift of the peak clearly shows the slowing down of the scattering process under stronger interactions. Interaction-dependent loss leads to degradation of contrast of the scattering patterns, and also results in deviations of population ratios from theoretical results (see text in Section 4.2). For $a_{dd} = 2526a_0$ at long pulse times ($t > \sim 600\mu s$), the images acquired have poor signal-to-noise ratio and population determination by fitting gives rather unclear results.

contrast and drives the evolution at a rate such that the effect of interaction, which becomes apparent at longer times, is clearly demonstrated. Similar to what is observed in the Raman-Nath regime, we find loss from the condensed component into the background which becomes more prominent for increased interaction strength. It can be seen that for $a_{dd} > 2500a_0$, the scattering pattern becomes barely recognizable from the image. We apply the same procedure as in the previous section to extract the momentum evolution. Due to loss by collisions between different momentum modes, the ratios between the populations are changed. As a result, the normalized populations at high interaction show deviations from numerical simulations of the scattering processes. We look into the interaction dependent loss in Section 4.2.

The time carpets in Figure 4 show a periodic time evolution of scattering patterns in momentum space, as expected for the channeling regime. Comparing the population evolution curves for different interaction strengths reveals a slowing down phenomenon of the diffraction process. It can be seen that the time point where the coherent population is restored to the $0\hbar k$ mode (recurrence of the mBEC) occurs at increasingly later time points, showing that the evolution becomes slower in the presence of stronger interaction.

# 4 Effects of interaction

## 4.1 Interaction induced slowing of scattering processes

To assist in understanding the effect of interaction on the population evolution, we performed Gross-Pitaevskii equation (GPE) simulations, which include mean field $s$-wave scattering between molecules (see details in Appendix A.3). As can be seen from Figure 4 showing the data for $U_0 = 50E_r$, the population evolution obtained by simulation is in good agreement with the experimental observations.

The slowing down of the scattering process at early times can be qualitatively understood to be associated with the formation of density grating across the condensate, which arises in the presence of the lattice potential. Initially, the lattice potential generates a phase modulation in the condensate, which leads to an emerging density grating. The $0\hbar k$ mode decreases while the higher momentum modes grow. The density profiles generated by the simulation demonstrating this process are shown in the inset of Figure 5. For convenient visualization, the simulation here is carried out with a lattice depth of $U_0 = 500E_r$ to obtain a rapid population evolution. An obvious density grating has already formed at approximately $2\,\mu s$. The effect of repulsive mean-field interaction, contrary to the optical lattice, tends to smooth out the density grating. The interaction counters the effect of the lattice, and hence slows down the population evolution during the scattering process.

Since the formation of density grating in the condensate is associated with the emergence and evolution of populations diffracted to higher momentum modes, one can also expect the effect of interaction would reverse the population evolution when the lattice pulse is turned off, and this is demonstrable with simulation. Figure 5(left) shows the population evolution of the $0\hbar k$ mode population obtained from the numerical simulation, in which a lattice potential with $500E_r$ of depth is pulsed onto a BEC under zero interaction. The lattice is then switched off at $3\,\mu s$ and later at $5\,\mu s$ interaction is turned on. It can be seen that with null interaction, the mode populations maintain the values from the moment the lattice is switched off. On the other hand, when the interaction is switched on, we observe the reversal of the population

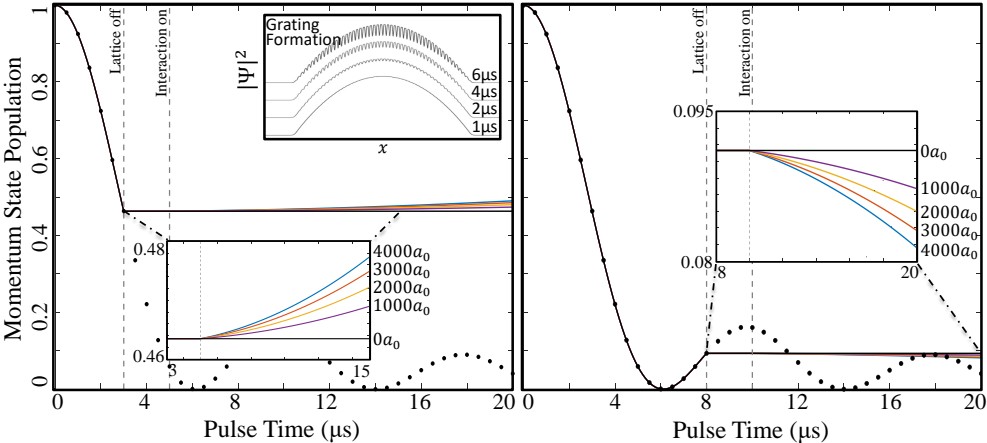

Figure 5: The simulated population evolution of $0\hbar k$ momentum modes with quenching interaction and $U_0 = 500E_r$. Solid lines show the simulation results and dots plot the Bessel function prediction. The two vertical dashed line marks the time of lattice switch-off and interaction switch-on, *(left)* $t_{\text{lat.off}} = 3\,\mu s$ and $t_{\text{int.on}} = 5\,\mu s$, *(right)* $t_{\text{lat.off}} = 8\,\mu s$ and $t_{\text{int.on}} = 10\,\mu s$ respectively. The inset shows the formation of density grating in the condensate wavefunction.

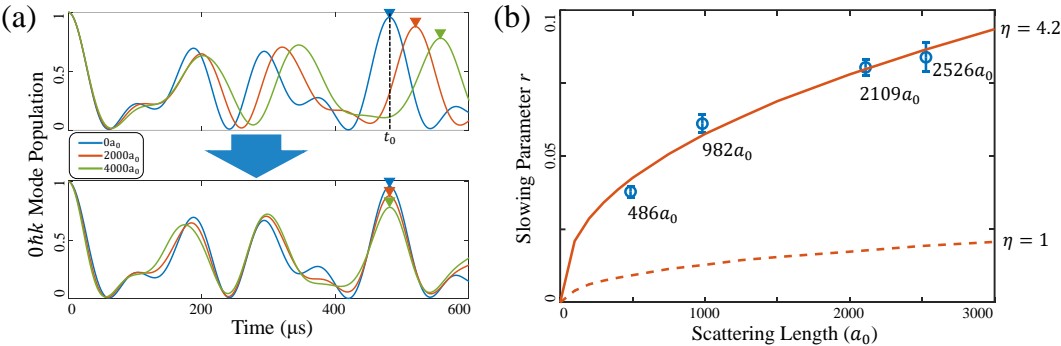

Figure 6: *(a)* 1D GPE calculation illustrating the slowing of the time evolution of the zeroth order diffraction peak with interaction strength. The curves can be aligned by re-scaling in time to match the first condensate recurrence points to the null inter-action case ($t \rightarrow t/(1+r)$). *(b)* The delay parameter $r$ inferred from experimental measurements (circles) in Figure 4 versus the dimer-dimer scattering length $a_{dd}$. Error bars generate with 10 repeated measurements. Solid line and dash line are produced by GPE simulation with $U_0 = 50E_r$ for $\eta = 4.2$ and $\eta = 1$, where $\eta$ is the phenomenological factor we included to account for additional contribution to interaction, $g = \eta g_{1D}$ (section A.3).

evolution due to scattering, consistent with what is predicted based on the physical picture. It is also evident from comparison that stronger interaction results in a faster reversal of the evolution.

At later times the density grating soon becomes complicated in structure (see Appendix A.3). It is then not obvious to draw a simple interpretation for the effect due to interaction, for instance at a time when the $0\hbar k$ mode population grows and the higher modes diminish. However, the 1D mean-field simulation demonstrates similarly, that the effect of interaction reverses the population evolution due to scattering (Figure 5(right)), again leading to faster reversal for stronger interactions.

In light of the physical picture and the tests with simulations, we expect the slowing of scattering processes to be dependent on interaction strength, and in addition, that the phenomenon can be captured by a 1D model [34]. Note from Figure 4 that the overall trend of the evolution, the features such as peaks and troughs of the curve, are maintained to a good degree, especially prior to the first recurrence of the $0\hbar k$ condensate. This allows us to further examine the phenomenon quantitatively with respect to the interaction strength, by identifying the peak locations to characterize the evolution.

In a simplified picture, the slowing effect results in a linear scaling of the population evolution in time by a factor $(1 + r)$. That is, assuming the slowing effect is uniform over time. As shown in Figure 6(a), we choose the first condensate recurrence point of the $0\hbar k$ condensate population in our measurement as the reference. We then use it to determine the time scaling factor $(1 + r(a))$ for a given $s$-wave scattering length $a$. The slowing phenomenon due to interaction countering the lattice potential can also be qualitatively seen as an effective reduction of the lattice depth by the factor $1/(1 + r(a))$, such as observed for a quasi-periodic lattice, where the onset of localization is shifted by repulsive interaction to deeper lattice potentials [35]. Figure 6(b) plots the values of $r(a)$ determined for the experimental data with $50E_r$ presented in Figure 4.

Mean-field 1D simulations indeed reproduce the slowing effect (see Appendix A.3 for details). However, simulations performed with experimental conditions and the scattering lengths at respective magnetic fields generate evolution with much weaker slowing down than

that observed experimentally. This indicates the presence of additional interaction energy contributing to the effect. Due to limited experimental knowledge of the in-trap loss mechanisms and the absence of accurate theoretical models, we consider here a phenomenological approach. Based on the above discussion, we therefore include a simple factor $\eta > 1$ in the mean-field simulation to account for the additional interaction via $a_{dd} \rightarrow \eta\, a_{dd}$. Using 3000 molecules for all cases in the simulation, in accordance with the initial condition of the experiments, we find that for a single value $\eta = 4.2$ the simulation generates results closely fitting the experimental observations, as shown in Figures 4 and 6. The value of $\eta$ independently determined for each field show slight fluctuations, but no dependence with respect to the scattering length is found.

The additional interaction energy could partly be accounted for by dissociation of molecules into atoms during the lattice pulse. The molecule to atom $s$-wave scattering length is given by $1.2a_{12}$ [36], which is double the value for dimer-dimer scattering. In addition, the dissociation of molecules leads to an increase in particle number, hence increasing the interaction strength.

We confirm the presence of free atoms following a 60 $\mu s$ lattice pulse by driving the $|2\rangle \rightarrow |3\rangle$ transition with a radio frequency (RF) pulse [37]. For sufficiently large binding energy and sufficiently long RF pulses we are able to distinguish between the transition from molecules and the transition from free atoms. For the corresponding RF frequency, a reduction of state $|2\rangle$ atoms detected by absorption confirms the presence of free atoms. Limited by the current low Rabi frequency, the RF pulse has to be performed in trap for several $ms$, leading to significant 3-body loss [38]. The method is hence at this point insufficient to provide a quantitative measurement. Also, due to the constraint on pulse length imposed by the requirement to resolve the bound-free and free-free transitions, we expect accurate characterization will be increasingly challenging close to the Feshbach resonance, where the molecule binding energy is small [39].

## 4.2 Condensate loss due to incoherent collisions

Both in the Raman-Nath regime and beyond, it is observed that the particle numbers in the condensed momentum peaks decrease with the increase of interaction strength, while the presence of background particles becomes enhanced.

In order to investigate at which stages and by what processes the loss occurs, we make use of a technique implementing a particular lattice pulse sequence [40]. By applying a designed lattice pulse sequence to a known initial distribution, the populations of the higher momentum modes can be 'returned' to the $0\hbar k$ mode as long as coherence is maintained. Here we choose the initial lattice pulse to be $t_0 = 60\,\mu s$ and $U_0 = 50E_r$. Figure 7(a) shows a qualitative sketch of the pulse sequence designed for this test, and the absorption images after the time-of-flight and focusing, at which point the different momentum modes are already well separated. We compare the cases where the cloud was released when (1) no pulse is applied, (2) the initial pulse is applied and only a very low population remains in the zero momentum mode, (3) the zero momentum mode is restored by the pulse sequence.

We can identify two distinct stages of loss. With the initial pulse (after $t_0$), most molecules will populate the $\pm 2\hbar k$ and $\pm 4\hbar k$ modes, while the background emerges with stronger interaction. The spherical structure and its '$2\hbar k$ radius' suggest that one of the major sources of the background is the collisions between the $\pm 2\hbar k$ modes [41, 42]. After being released from the dipole trap, it takes $\sim 6\,ms$ for different momentum modes to separate in space. The recoil momentum $2\hbar k$ is significantly larger than the superfluid critical velocity [43]. Collisions resulting from relative movements lead to decoherence and redistribution of momenta, which are enhanced under higher interaction. With the additional pulse sequence applied (after $t_4$), the coherent population is returned to the zero momentum mode, and the contribution of col-

lisional loss during mode separation is suppressed. Although the BEC can be mostly restored with the pulse sequence, an interaction dependent loss can still be observed. We attribute this to in-trap loss which occurs during the lattice pulse. The presence of unbound atoms confirmed by RF spectroscopy is also consistent with loss occurring during the pulse. For case (3) there may be extra loss due to the additional pulses. However, this would shift the line of (3) upwards and decrease the difference between cases (2) and (3), leading to an underestimate of the collisional loss. Therefore the difference between the cases (2) and (3) unambiguously demonstrates collisional loss between the momentum modes during the course of separation.

Figure 7(b) shows the total loss at different stages and interactions. When determining the population of condensate peaks and background using the fitting algorithm, the calculation always gives a non-zero background number even for a pure BEC. This error is influenced by the imaging noise, since the fitting always includes part of the noise as the fitted distribution. So we used the result obtained with the BEC as a reference for the error level. We can identify the difference between measurements "after $t_0$" and "after $t_4$" as the contribution of collision loss after release from the dipole trap, and the difference between "after $t_4$" and the error level as the contribution of molecule disassociation after the initial $60\mu s$ lattice pulse.

We simulated the collision loss during the course of separation by calculating the collision events between each two momentum groups with a simple model (for details see Appendix A.4). The calculation for each scattering length and pulse time is initiated with the mode occupations obtained from the corresponding GPE simulation, subsequently corrected for the (early stage) in-trap loss within the first $60~\mu s$ as has been experimentally characterized (shown in Figure 7). The later loss during the lattice pulse (in-trap) is difficult to model and is not included in this calculation. Expansion following trap release is then calculated by the model, giving quantitative estimates of collision loss during the separation between the momentum modes (Figure 8 and 9). From the simulation we obtain the estimated total remaining condensed population in the momentum modes, as well as the remaining population

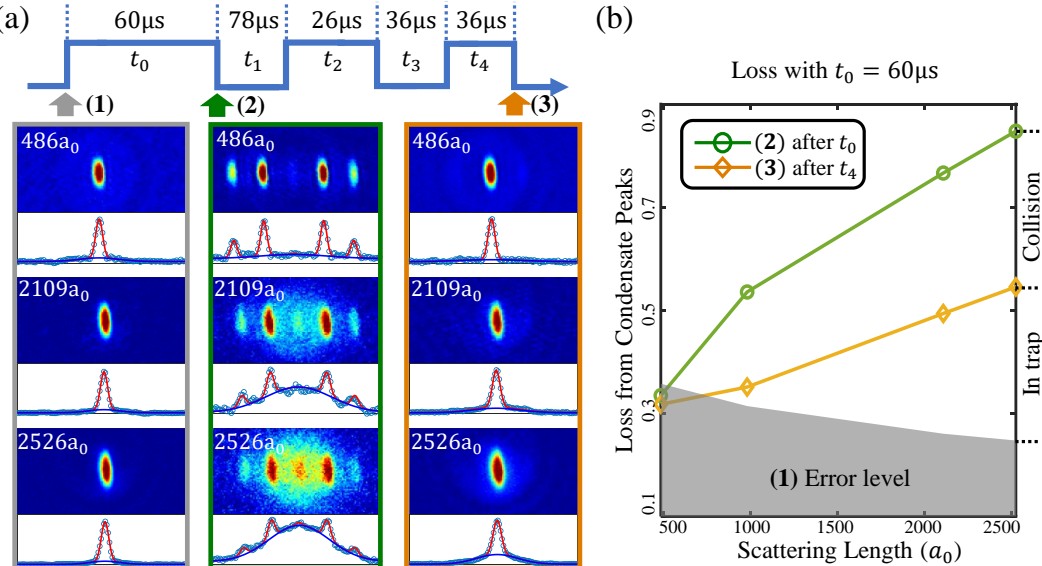

Figure 7: *(a)* Schematic of the pulse sequence and the corresponding absorption images at different interactions. $[t_0, t_1, t_2, t_3, t_4] = [60, 78, 26, 36, 36]\,\mu s$ for $U_0 = 50E_r$. *(b)* Calculated loss plot against scattering length at different stage. The error level is calculated from the BEC images (1). Difference between line (3) and error level is the loss during the lattice pulse. Difference between line (2) and (3) is the collision loss during TOF.

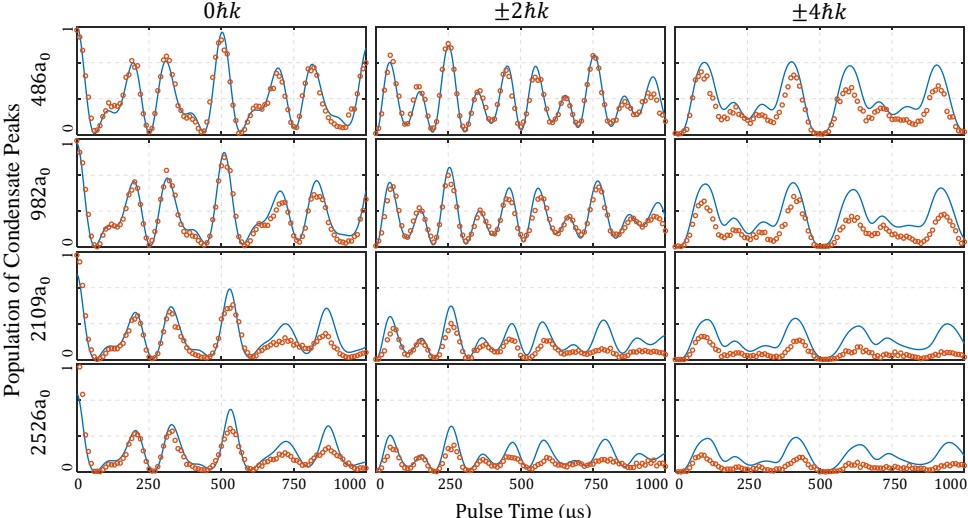

Figure 8: The population evolution of $0\hbar k$, $\pm 2\hbar k$, $\pm 4\hbar k$ modes at different interaction strengths (open symbols) with collision simulation results (solid line) for $U_0 = 50E_r$. The initial condition of the simulation is a product of the GPE calculation multiply by the loss factor calculated from $t = 60\mu s$ (Figure 7). Different from Figure 4(right), here we normalize to the initial condensate number.

in the individual modes. The results show reasonable agreement with the experimentally observed $0\hbar k$ mode for short pulse times, and larger deviation for long pulse time, indicating the increasing contribution from in-trap loss over time. Additionally, the losses from higher modes ($\pm 2\hbar k$,$\pm 4\hbar k$) are increasingly underestimated (Figure 8), which we attribute to the approximations taken by the scattering model. In particular secondary collisions or corrections to the scattering cross-section beyond $s$-wave scattering are not taken into account. Quantifying secondary collisions is difficult given our experimental scenario. Due to the axial length of the condensate in our experiment, the momentum modes mutually separate over the course of several milliseconds. If the collisions occur in a small and well-defined range of space and time, one can expect to observe clean $s$-wave collision halos, and deviations from the expected profile would then indicate additional processes such as secondary collisions [44]. This is not the case for our experiment. Our situation is further complicated by the presence of loss caused by the lattice pulse.

The deviations of the normalized populations observed experimentally (Figures 3 and 4) from the GPE simulation results can be understood with the collision process during the separation of momentum modes. Consider the molecules belonging to two velocity groups moving relative to each other and colliding. Assuming the colliding molecules are immediately removed from the coherent condensates and hence secondary collisions can be neglected, it follows that the two groups incur the same number of particle losses to each other. When the initial populations are not equal, the group with higher population will have an increased portion of the remaining coherent particles after collisions have taken place. Hence the more highly populated modes tend to increase in ratio. From Figure 3 ($2109a_0$) and Figure 4 such features can be seen. Under strong interactions, when the $0\hbar k$ mode population is high (peaks), the normalized population observed experimentally ends up occupying a higher portion of the total remaining condensed molecules than the simulation result. The features become more prominent for stronger interactions and hence stronger losses.

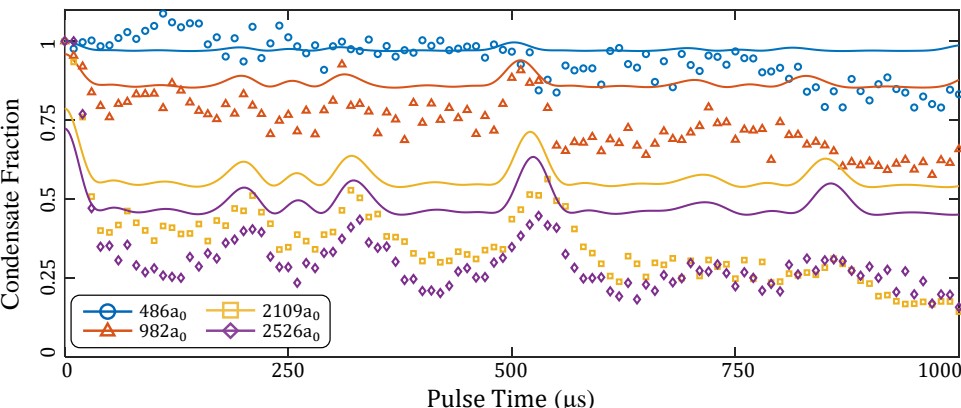

Figure 9: The condensate fraction at different interaction strengths (open symbols) with collision simulation results (solid line) for $U_0 = 50 E_r$. The increasing discrepancies between calculated results and experimental data indicate the continuing in-trap loss over time.

# 5  Conclusions and outlook

Diffraction of strongly interacting matter waves closely follows the well studied non-interacting single particle case. The main modifications of the observed diffraction come either from (i) incoherent processes like final state interactions which leads to a broad scattering background, and Feshbach molecule dissociation, and (ii) a coherent process initiated by the self interaction of the evolving matter wave interference patterns inside the standing light wave. Investigations for the possible physical processes leading to strengthened interaction are currently underway.

We verified the dominance of coherent processes during the diffraction dynamics in the standing wave grating by applying additional pulses which allowed to recombine the diffracted orders back into the zero momentum mode by constructive interference.

Going beyond the present work it would be interesting to study (i) the regime where the emerging diffraction orders are all safely below the critical super-fluid velocity; (ii) the influence of the switching on and switching off of the standing wave; (iii) Bragg diffraction and the modification of its intricate wave fields inside the light crystal [45, 46] by the strong interactions and (iv) in addition strong interactions should also lead to squeezing and entanglement between the diffracted beams. We expect the latter to be relevant for interferometry, where the squeezing and entangled created by strong interaction may be desired for meteorological advantage [21, 47]. Further, demonstrating the influence of interactions on the performance of a full multi-mode interferometer sequence [48] with lithium Feshbach molecules is an interesting avenue for further investigations.

# Acknowledgements

This research was supported by the DFG-SFB 1225 'ISOQUANT', with the Vienna participation financed by the by the Austrian Science Fund (FWF)(grant number I3010-N27), and the Wiener Wissenschafts- und Technologiefonds (WWTF) project No MA16-066 (SEQUEX). S.E. aknowledges an ESQ (Erwin Schrödinger Center for Quantum Science and Technology) fellowship funded through the European Union's Horizon 2020 research and innovation programme under the Marie Skłodowska-Curie grant agreement No 801110. This project reflects only the author's view, the EU Agency is not responsible for any use that may be made of the informa-

tion it contains. ESQ has received funding from the Austrian Federal Ministry of Education, Science and Research (BMBWF).

# A   Appendix

## A.1   Experimental apparatus

The layout of the experimental apparatus is shown in Figure 10. Lithium atoms from an oven going through a Zeeman slower are laser cooled and collected in a magneto-optical trap (MOT) at the metal chamber using the $D_2$ optical transition ($2S_{1/2} \rightarrow 2P_{3/2}$). The MOT consists of the cooling and repump light which excite atoms from the $F = 3/2$ and the $F = 1/2$ states, respectively, and typically collects $2 \times 10^8$ atoms at $\sim 1$ mK. Following compression by ramping the laser frequency close to resonance and decreasing optical intensity, the temperature is reduced to $330 \, \mu$K, while keeping $1 \times 10^8$ atoms.

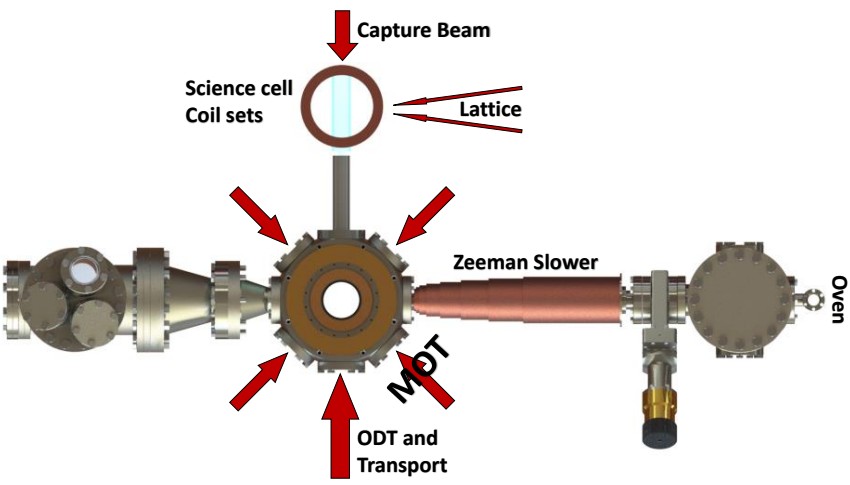

Figure 10: Experimental apparatus.

The atoms are subsequently transferred to an optical dipole trap (ODT) formed by 1070 nm laser from a high power Ytterbium fiber laser (IPG YLR-200-LP-WC). By extinguishing the repump light $100 \mu s$ earlier than the cooling light, the atoms are pumped to the $F = 1/2$ states, which are the lowest two magnetic sublevels $|1\rangle$ and $|2\rangle$.

After transferring to the optical dipole trap, the quadrupole magnetic field of the MOT is switched off and a uniform offset magnetic field provided by the Feshbach coil pair is switched on. In the presence of a magnetic field of 540 G, radio frequency (RF) sweeps mix the two states to ensure balanced populations for efficient evaporative cooling.

Evaporative cooling is then carried out by decreasing the optical power of the dipole trap, the first evaporative cooling stage performed at the MOT chamber is done under a magnetic field offset of 780G, giving strong interaction between the spin states and hence rapid thermalization.

Evaporation at the MOT chamber is performed without reaching quantum degeneracy. The atoms are then transferred to the transport beam shaped by a two-lens setup including a tunable lens (Optotune EL-16-40-TC-NIR-20). The beam focus position of the transport trap varies with the focal length of the tunable lens. The cloud moves with the trap focus to the science cell and is then transferred to the capture beam at the cell. Final evaporation is carried out in the capture beam to prepare the sample for the experiment, producing a degenerate

Fermi gas or mBEC.

The lattice setup is based on an equal-path interferometer design that minimizes the path length difference to provide high passive phase stability. An optical lattice formed by two coherent laser beams of wavelength $\lambda = 1064$ nm crossing with an angle $\theta = 15°$ gives lattice spacing $D = \lambda/[2\sin(\theta/2)] \approx 4\,\mu$m. The lattice depth was determined for the $500\,E_r$ case where the population evolution is fast, and therefore the other effects are insignificant. By fitting the measured zero momentum population evolution with the Bessel function and determining the time of the first minimum, the lattice depth is inferred.

## A.2 Pulse sequence design

As illustrated in Figure 7(a), the momentum mode populations are accurately controlled with lattice pulses. A single lattice pulse transfers a BEC initially in the zero momentum mode into high momentum modes. Subsequently, another two lattice pulses can bring most particles back to the $0^{\text{th}}$ mode prior to TOF. To characterize the collisional loss effect in different experimental stages, we compare two cases, (i) the cloud is released from the capture beam with multiple momentum modes, which separate and penetrate through each other during TOF, (ii) the cloud is released with high momentum modes being eliminated, hence strongly suppressing the collisions during expansion.

The pulse sequence is designed with particular pulse durations and intervals for a given lattice depth $U_0$, aiming at maximizing the overlap between the final wavefunction and the BEC wavefunction in the $0^{\text{th}}$ mode [40]. Supposing that $\psi_0$ is the state of a BEC, we calculate the Bloch state after applying a pulse sequence $[t_0, t_1, t_2, t_3, t_4]$ (see Figure 7a),

$$|\psi_{final}\rangle = \prod_{j=4}^{0} \hat{\mathcal{U}}_j |\psi_0\rangle, \tag{1}$$

where $\hat{\mathcal{U}}_j = e^{-i[\hat{p}_x^2/(2m) + U_j \cos^2(kx)]t_j/\hbar}$ is the evolution operator in the $j^{\text{th}}$ step. The interaction term is neglected because the pulse durations are smaller than the timescale of which the slowing effect appears significantly. The duration of the initial pulse $t_0$ is fixed at $60\,\mu s$ so that it covers the strong loss region observed in Figure 9. The potential depth $U_j$ is set to $U_0$ and 0 during the pulses and the time intervals, respectively. $U_0$ keeps constant for all three pulses. The time sequence is determined by maximizing $|\langle\psi_0|\psi_{final}\rangle|^2$. For the parameters we choose, $[t_0, t_1, t_2, t_3, t_4] = [60, 78, 26, 36, 36]\,\mu s$ and $U_0 = 50E_r$, leading to $|\langle\psi_0|\psi_{final}\rangle|^2 = 0.94$.

## A.3 GPE simulation

We performed mean-field simulations based on the Gross-Pitaevskii equation (GPE),

$$i\hbar\partial_t\Psi = \left[-\frac{\hbar^2}{2m}\partial_x^2 + \frac{1}{2}m\omega_x^2 x^2 + U(x) + g_{1D}|\Psi|^2\right]\Psi, \tag{2}$$

to investigate the slowing down effect due to interaction. Here $m$ is the mass of a molecule, $g_{1D} \sim a_s$ the effective 1D interaction constant (see below), $\omega_x$ is the harmonic trapping frequency of the capture beam. The lattice potential $U(x) = U_0 \cos^2(\pi x/D)$ has a lattice period $D = 4\mu$m, setting the recoil energy $E_r = \hbar^2 k^2/2m \approx 250$ Hz with $k = \pi/D$. We implemented (i) a homogeneous 1D simulation, and (ii) a 1D simulation with settings corresponding to the experimental conditions.

To demonstrate the phenomenon and to test the physical picture (see section 4.1), we simulated a homogeneous 1D system, for which the interaction strength is fully characterized by the chemical potential $\mu = g_{1D}n_{1D}$, where $n_{1D}$ is the particle density and $g_{1D}$ the 1D interaction parameter. The simulations were performed for 25 lattice periods with 50 spatial

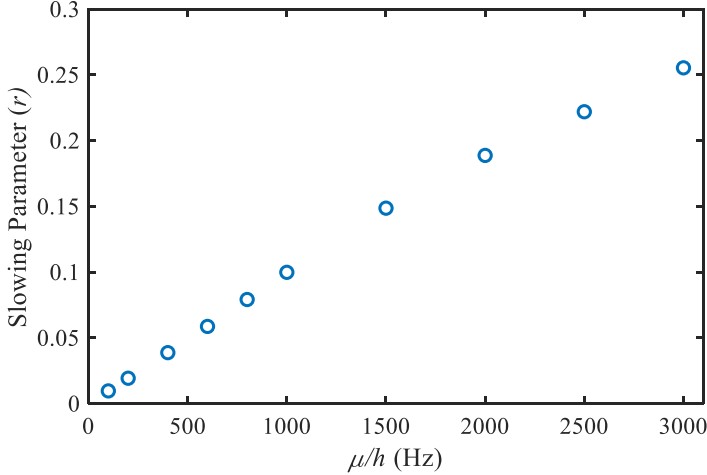

Figure 11: 1D GPE simulation demonstrates the slowing down effect due to interaction. The simulation is performed with a lattice potential depth of 50 $E_r$. The slowing parameter $r$ is found to monotonically increase with in interaction strength and is approximately proportional to the chemical potential for weak interaction.

points per lattice period (80 nm resolution) and 100 ns time steps. The 1D simulation clearly demonstrates the slowing effect, as plotted in Figure 11. For each interaction strength the slowing parameter $r$ is determined by comparing the condensate recurrence time to the null interaction case. The phenomenon is found to be stronger with increasing chemical potential.

In order to perform the simulations that correspond to the experiment, we measured the trap frequencies and the in situ size of the mBEC for each interaction strength. Figure 12 shows an example absorption image of our mBEC, and the comparison of the fitted condensate sizes to the theoretical Thomas-Fermi radii without free parameters. The experimental conditions determined are used for the GPE simulation.

To set up a 1D simulation corresponding to the experimental conditions, we integrate out the radial directions of the 3D Thomas-Fermi profile of the mBEC to obtain the effective 1D in-

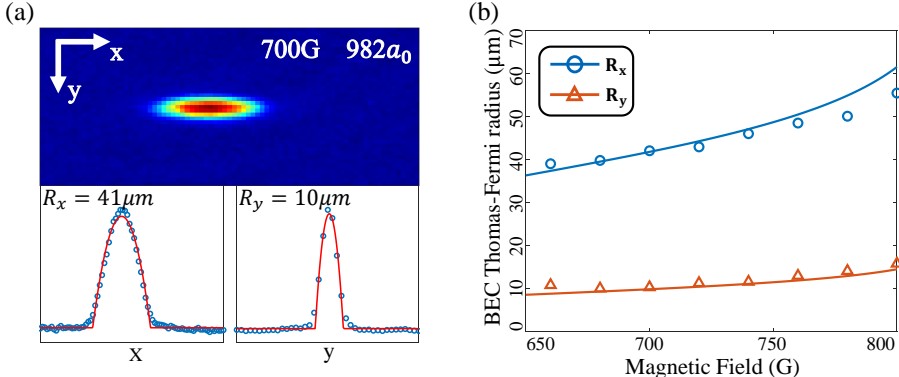

Figure 12: *(a)* In situ profiles of the mBECs at 700G and *(b)* the measurement of Thomas-Fermi radius (open symbols) with calculated result (solid line). Trap frequencies $(f_x, f_y, f_z) = (16, 74, 68)\,\text{Hz}$ and molecule number 3000 are used for the calculation. The results agree well with theoretical prediction based on the measured molecule number and trap frequencies.

teraction parameter, $g_{1D} = 16\hbar^2 a_{dd}/(3mR^2_{TF_r})$. The simulations were carried out with 10 nm spatial resolution and 100 ns time steps. The spatial resolution is determined by the requirement of Fourier space width in the case of Raman-Nath regime, involving higher momenta. The 1D simulation has been checked against a full 3D simulation, confirming that radial excitations have minor influence on the evolution within the timescales considered in this work. The 1D simulation is then used to perform the calculation shown in Figure 5 to check our proposed physical picture for the slowing phenomenon. For comparison with experimental data, we include a fudge factor $\eta > 1$ to account for additional contribution to interaction. This may include, but is not limited to, the effect of unbound atoms from dissociation of Feshbach molecules (see main text in section 4.1).

### A.4  Collision simulation

We simulate the incoherent collision loss by calculating the expected number of collision events during the separation of the momentum modes during time-of-flight.

The simulation was carried out in a simple setting, taking the molecules of each momentum mode to be uniformly distributed within a cylinder, with the half-length and radius given by the axial and radial Thomas-Fermi radii of the mBEC. The expected collision events encountered by one particle over a distance of travel is given by the number of particles in the other momentum group contained in the cylinder defined by the displacement of the particle and its scattering cross-section.

Due to symmetry of this setting, the numerical calculation can be done in 1D, where the line density of each momentum group in an array is evolved. In each step the groups with momentum difference $2\hbar k$ shift in relative position by the distance of one cell. For two cells that move across each other, the estimated number of collisions is given by $N = (n_1 \pi R^2 dx) n_2 A dx = n_1 n_2 \pi R^2 A dx^2$, where $n_i$ are the (3D) particle density of the cells, $R$ the Thomas-Fermi radius of the mBEC, $A = 8\pi a_s^2$ the collision cross-section, and $dx$ the width of the cell, which is also the distance of relative travel between the two cells. The line density of a cell is $n_i \pi R^2$, and decreases by $N/dx$ in this calculation step. We take $dx$ to be sufficiently small so that the collision event encountered by each molecule in one calculation step is less than unity. The procedure obtains the expectation value of the number of collision events, and therefore the decrease in particle number from both cells.

The numerical calculation takes into account the $0\hbar k$, $\pm 2\hbar k$, and $\pm 4\hbar k$ momentum modes, which have significant occupations during the scattering process. The molecules to which collisions occur are considered to be lost from the coherent condensate. Hence in the next step of the calculation the density is decreased accordingly. The lost molecules are taken to be immediately removed from the cloud. Secondary or further collisions are supposed to be rare and not taken into account.

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
