# Peer review of "Diffraction of strongly interacting molecular Bose-Einstein condensate from standing wave light pulses"

_SciPost Physics, doi:SciPost Phys. 12, 154 (2022)_

## Round 1 · Referee Report · Anonymous (Referee 1) · 2022-2-7

Strengths

1- Novel experimental results in a simple, yet rich physics setting 2- Thorough theoretical analysis 3- Excellently written, very clear figures 4- Additional experiments (pulse sequences) were designed to answer open questions

Weaknesses

1- Using a molecular BEC instead of a standard bosonic Feshbach-able species slightly complicates the interpretation of the measured results. 2- The measurements do not go beyond the mean-field regime.

Report

The authors perform an in-depth experimental study of the effects of mean-field interactions on Kapitza-Dirac diffraction. The diffracting matter waves resulting from sudden flashes of an optical lattice consist of a molecular Bose-Einstein condensate of fermionic lithium-6. On the one hand, the authors study the transient, short-pulse regime. Except for a broadened background in time-of-flight, no major signatures of the interactions were observed. On the other hand, longer pulses (up to 1ms) lead to an interesting 'slowing' effect of the diffraction dynamics in addition to the previously observed incoherent background.

The slowing effect is qualitatively captured by Gross-Pitaevskii simulations. To my understanding, it results from the repulsive mean-field potential of the condensate counteracting the lattice potential and effectively reducing the lattice depth. The magnitude of the slowing effect is roughly a factor of four stronger than expected from the numerics and the authors provide possible reasons for this discrepancy.

The authors proceed to design experiments to investigate a possible cause of the incoherent background in time-of-flight expansion. A suitable lattice pulse sequence is designed to rephase all particles into the zero momentum peak, suggesting that the broad background occurs during the expansion. Overall, the authors provide convincing theoretical and experimental arguments as to why the incoherent background occurs.

To summarise, the authors provide novel experimental measurements of Kapitza-Dirac diffraction in the interacting regime, which has largely remained unexplored to date. Therefore, I would recommend publication in SciPost Physics.

Requested changes

1- The authors provide a nice qualitative explanation for the slowing effect (section 4.1 on page 7), as reduction of lattice depth due to a repulsive mean-field potential. The argument could be made even more explicit by calculating an effective reduction of lattice depth in recoil energies. This could be achieved by fitting the diffraction dynamics to the noninteracting numerics, leaving the lattice depth as a free parameter. Then the y-axis in Fig.6b could be expressed as 'cancelling lattice depth' (if the relation is linear in this regime).

2- A few technical details on the optical lattice are missing: How is the lattice depth calibrated? What is the estimated phase stability of the lattice? Could a jittering optical lattice (due to an unstable phase) contribute to the interaction-dependent incoherent background observed in the experiment?

3- As far as I understood, the numerics are scaled on the x-axis to match the experimental data, taking the slowing parameter r as a free parameter. Throughout the plots in the manuscript, the authors should clarify which theory curves are without free parameters and which curves are fits to the data (especially in Fig. 4 on the right, in which the theory agrees very well with the data - is this a fit or an a priori theory curve without free parameters?).

4- What is expected to happen in the beyond-mean-field regime?

5- The effects of mean-field interactions on rapid triangular optical lattice ramps were previously studied in "Observing Localization in a 2D Quasicrystalline Optical Lattice" by Sbroscia et al. PRL 125, 200604 (2020). Can the authors comment on the relation to their results?

Smaller changes: - Why is the lattice depth changed from 500Er to 50Er between sections 3 and 4? The authors should explain this briefly. - a typo at the bottom of page 3: "afterwords" - Fig 4 (right): the x-label "1000" does not match the tick of 1000us. - Fig 4: the slowing is not evident from the dashed line as it is. Stronger tick markings (and more ticks) or a grid could help, or maybe a zoom on the relevant region with the dashed line (such as Fig. 6a). - In general, the ticks in all figures except Fig6b are too thin.

  • validity: top
  • significance: ok
  • originality: high
  • clarity: high
  • formatting: excellent
  • grammar: excellent

Author:  Chen Li  on 2022-04-01  [id 2346]

(in reply to Report 1 on 2022-02-07)
Category:
answer to question

We thank the Referee for the insightful comments and the constructive suggestions. Below we provide a detailed response to all points raised by the Referee. In the “weaknesses” section, the referee pointed out that ‘Using a molecular BEC instead of a standard bosonic Feshbach-able species slightly complicates the interpretation of the measured results.’ The referee is correct that, the breaking up of molecules results in a more complicated situation and leads to the difficulty in the quantitative analysis of the observed results. However, using $^6$Li$_2$ molecules is to our knowledge the only option to prepare an in-equilibrium strongly interacting state as the starting point of the experiment. This is because inelastic processes are strongly suppressed due to the fermionic nature of the $^6$Li atoms, providing a long lifetime in the strongly interacting regime. The following two references demonstrated the relevant microscopic physics: Petrov et al. PRL 93, 090404 (2004), and Petrov et al., PRA 71, 012708 (2005). We clarify this point in the revised draft.

Requested Changes Q1. The authors provide a nice qualitative explanation for the slowing effect (section 4.1 on page 7), as reduction of lattice depth due to a repulsive mean-field potential. The argument could be made even more explicit by calculating an effective reduction of lattice depth in recoil energies. This could be achieved by fitting the diffraction dynamics to the noninteracting numerics, leaving the lattice depth as a free parameter. Then the y-axis in Fig.6b could be expressed as 'cancelling lattice depth' (if the relation is linear in this regime). A1. As the referee correctly pointed out, the slowing effect can indeed be qualitatively seen as effectively reducing the lattice depth, by the factor $1/(1+r)$. We have included this alternative picture in the revised text. This is also related to the observation of interaction effects in lattices reported in PRL 125, 200604 (2020), as pointed out by the referee in Q5.

Q2. A few technical details on the optical lattice are missing: How is the lattice depth calibrated? What is the estimated phase stability of the lattice? Could a jittering optical lattice (due to an unstable phase) contribute to the interaction-dependent incoherent background observed in the experiment? A2. The lattice depth was determined for the 500 $E_r$ case where the population evolution is fast and therefore the other effects are insignificant. By fitting the measured zero momentum population evolution with Bessel function and determining the time of the first minimum, the lattice depth is inferred. We clarify these technical details in the Appendix of the revised manuscript. Regarding the lattice stability and the possibility of lattice jittering contributing to the incoherent background, we note the following. - Fig.7 shows that the incoherent background diminishes, when decreasing the interaction. This shows that the process creating the incoherent background is predominantly interaction dependent. - We measured the mechanical vibrations and found that they are (i) predominantly low frequency and (ii) smaller than 5% of the lattice spacing, which leads us to conclude that vibrations are not important.

Q3. As far as I understood, the numerics are scaled on the x-axis to match the experimental data, taking the slowing parameter r as a free parameter. Throughout the plots in the manuscript, the authors should clarify which theory curves are without free parameters and which curves are fits to the data (especially in Fig. 4 on the right, in which the theory agrees very well with the data - is this a fit or an a priori theory curve without free parameters?). A3. The solid curves in fig.3 are given by Bessel functions calculated according to the calibrated lattice intensity without any free parameters. The solid curves in Fig.4 (right) are produced by 1D mean field simulations with interaction, including an interaction scaling factor $\eta$ shared by all simulation cases, to take into account the additional interaction energy, as explained in section 4.1. The time axes of these curves are not scaled. The slowing parameter $r$ is determined separately by scaling the time axis of the numerical simulation result for the null-interaction case to match the restoration of the zero-momentum condensate with that of the experimentally observed population evolution. We clarify these points in the revised manuscript with more precise statements.

Q4. What is expected to happen in the beyond-mean-field regime? A4. One would expect that in the beyond-mean-field regime, local fluctuations are not negligible and have to be taken into account. One result of these fluctuations would be a random phase shift at each lattice site. This would lead to shot-to-shot fluctuation of the momentum mode populations, which should in principle be detectable in experiments. Our experimental conditions are not in this regime. The chemical potential would set a very loose upper limit to these fluctuations. In our molecular condensates it ranged from ~180 Hz to ~350 Hz. Assuming the fluctuations be in as large as 10% of the chemical potential then even for the longest interaction time the resulting phase fluctuations would be well below 100mrad. Taking into account ensemble averaging over the 25 lattice sites that are illuminated in the diffraction experiments beyond mean field effects coming from local fluctuations will be very small.

Q5. The effects of mean-field interactions on rapid triangular optical lattice ramps were previously studied in "Observing Localization in a 2D Quasicrystalline Optical Lattice" by Sbroscia et al. PRL 125, 200604 (2020). Can the authors comment on the relation to their results? A5. We thank the referee to point us to the paper, which discusses a closely related phenomenon to our time delay, and provides a different view point on this. We address this in the response to Q1 and in the modification of the paper.

Q6. Why is the lattice depth changed from 500Er to 50Er between sections 3 and 4? The authors should explain this briefly. A6. We show the results for 500 $E_r$ in Sec. 3.1 and that for 50 $E_r$ in Sec. 3.2. The lattice depths were chosen in each case to enter the specific regimes. In Sec. 3.1, we study the diffraction within the Raman-Nath regime, where the particles remain approximately stationary during the lattice pulse. The lattice depth must be large so that the significant phase imprinting occurs in short time. While in Sec. 3.2, we study physics beyond the Raman-Nath regime. A weaker lattice restricts the particles within the first five momentum modes to achieve a decent imaging contrast and drives the evolution at a rate such that the effect of interaction, which becomes apparent at longer times, is clearly demonstrated. We clarify this the revised text in Sec. 3.2. In Sec. 4 we present more detailed analyses and discussions for the observed effects of interaction. The simulation test shown in Fig. 5 is done with 500 $E_r$ to obtain a more rapid evolution for easier visualisation, demonstrating the effect of interaction on population evolution. However experimentally we do not have sufficient signal to noise ratio to identify the slowing down in a short time. The experimental observation of the slowing down is only verified with 50 $E_r$. We clarify this in the revised text in Sec. 4.1.

Q7&Q8. a typo at the bottom of page 3: "afterwords". Fig 4 (right): the x-label "1000" does not match the tick of 1000us. A7&A8. We corrected the typo and proofread the manuscript again for any remaining errors.

Q9. Fig 4: the slowing is not evident from the dashed line as it is. Stronger tick markings (and more ticks) or a grid could help, or maybe a zoom on the relevant region with the dashed line (such as Fig. 6a). A9. We improved Fig. 4 by including a zoomed view.

Q10. In general, the ticks in all figures except Fig6b are too thin. A10. We improved all figures according to the referee’s suggestions.

---

## Round 1 · Referee Report · Anonymous (Referee 2) · 2022-3-14

Strengths

1- Novel experimental results 2- Strong interaction and long-pulse regimes explored 3- Theoretical modeling and interpretation of experimental observations 4- First experimental results on molecular BECs

Weaknesses

1- Several effect like dissociation of Feshbach molecules and multiple scattering are essentially uncontrolled and complicate the analysis 2- Interpretation of simulation results and discussion of the discrepancies not entirely satisfactory 3- No attempt was undertaken to model the effects of multiple scattering or three-body recombination

Report

This is an interesting work that pushes the widely used technique of diffracting matter waves from a standing light wave into new regimes by examining a molecular Bose-Einstein condensate and considering long pulses. It thus opens a new pathway in an existing research direction and should lead to follow-up work that would aim at better understanding the combined effects of molecular dissociation, three-body losses, and elastic scattering in molecular BECs.

I am not completely convinced by the discussion of interaction effects and comparison with Gross-Pitaevskii simulations. A slowing-down effect in the dynamics identified by the recurrence time of the zero momentum peak is identified in Gross-Pitaevskii simulations. In order to achieve qualitative agreement with experimental data, however, the rescaling parameter has to be multiplied with a fudge factor of value 4.2. It is argued that the mean-field simulations reproduce the experimentally observed effect and some qualitative justification of the fudge factor are presented. I do not find the arguments given entirely convincing and would lean to a different conclusion - namely that interaction effects are not satisfactorily captured by the 1D mean-field simulation.

The main argument given for the enhanced slowing down compared to mean-field simulations is the possibility of dissociated atoms being present in the BEC, while only molecules were assumed to be present for the simulation. As the molecule-molecule scattering length is 0.6a, where a is the atom-atom scattering length, and the atom-molecule scattering length is 1.2a, the presence of dissociated atoms may lead to increased mean-field interaction confounded by the fact that the number of interacting particles is also increased. Unfortunately, the presence of dissociated atoms, while confirmed in principle, could not be reliably quantified in the experiments.

I agree that the presence of dissociated atoms could increase effective mean field interactions, but would assume that such an effect would be quite moderate. I would expect an increase of the slowing parameter by less than a factor of 2, given that Fig. 6(b) seems to show something like a square root dependence of the slowing parameter on the interaction strength. I would rather interpret the bulk of the factor 4.2 increase of the slowing parameter as unexplained by the numerical mean-field simulation.

Loss of atom numbers is seen in experiments, but not quantitatively found in agreement with scattering calculations. Additional effects like three-body recombination and multiple scattering are believed to be present, and could in principle be modeled numerically as well. Have the authors considered at least estimating the quantitative effects of these processes?

Requested changes

1- The authors may want to consider mentioning in the abstract/conclusions that mean-field simulations and estimated effects of dissociated atoms cannot quantitatively explain the observed slowing-down effects of the dynamics. 2- Regarding the effects of multiple/secondary scattering events, the authors may want to consider estimating the expected effects, and refer to Thomas et al. Nat. Comms. 7, 12069 (2016) where such effects were quantified in experiments and simulation. 3- Fix typo "mater" in abstract. 4- Fix the first sentence of the second paragraph in Sec. 1 "The effect ...", as it's grammar appears to be broken. 5- The factor \eta should be defined in the caption of Fig. 6 for easier readability. 6- In the caption of Fig. 7 the difference between lines (2) and (3) is attributed to collision loss during TOF. Is this the only interpretation? Maybe a short discussion is warranted. 7- In the appendix A.3 on GPE simulations I would like to see, for completeness, more details that make it easier for others to reproduce the results: 7a- The formula used for g_1D should be given. 7b- The form of the external potential used (and relation to E_r) should be given (or a relevant reference cited). 7c- The chemical potential \mu shown in Fig. 11 should be defined in terms of the quantities of Eq. (1) (or should it appear in the equation?). 7d- Parameters of the simulation (how many lattice sites per computational box, discretisation parameters of the GPE) should be added.

  • validity: high
  • significance: good
  • originality: high
  • clarity: high
  • formatting: perfect
  • grammar: perfect

Author:  Chen Li  on 2022-04-01  [id 2347]

(in reply to Report 2 on 2022-03-14)

We thank the Referee for her / his constructive comments. The points raised by the Referee helped us to make our manuscript more comprehensive. Below we provide a detailed response.
In the “weaknesses” section, the referee pointed out that ‘Several effect like dissociation of Feshbach molecules and multiple scattering are essentially uncontrolled and complicate the analysis.’
The referee is correct that, the breaking up of molecules results in a more complicated situation and leads to the difficulty in the quantitative analysis of the observed results.
However, using $^6$Li$_2$ molecules is to our knowledge the only option to prepare an in-equilibrium strongly interacting state as the starting point of the experiment. This is because inelastic processes are strongly suppressed due to the fermionic nature of the $^6$Li atoms, providing a long lifetime in the strongly interacting regime. The following two references demonstrated the relevant microscopic physics. Petrov et al. PRL 93, 090404 (2004), and Petrov et al., PRA 71, 012708 (2005). We clarify this point in the revised draft.

Requested changes
Q1. The authors may want to consider mentioning in the abstract/conclusions that mean-field simulations and estimated effects of dissociated atoms cannot quantitatively explain the observed slowing-down effects of the dynamics.
A1. We agree with the referee's suggestion and clarify this point, which is already discussed in detail in section 4.1, now also in the abstract of the revised manuscript.

Q2. Regarding the effects of multiple/secondary scattering events, the authors may want to consider estimating the expected effects, and refer to Thomas et al. Nat. Comms. 7, 12069 (2016) where such effects were quantified in experiments and simulation.
A2. The referee raised a good point of consideration. The incoherent background we observed shows a spherical shape as expected for s-wave collision, however not a clear halo structure. Due to the axial length of the condensate in our experiment, the momentum modes take several milliseconds (depending on the scattering length) to mutually separate, which is not negligible compare to the total time-of-flight of 13 ms. If the collisions occur in a small and well-defined range of space and time, one can expect to observe clean s-wave collision halos, and deviations from the expected profile would then indicate additional processes such as secondary collisions, like identified in Nat. Comms. 7, 12069 (2016). This is not the case in our experiment. We add a short discussion in section 4.2 and cite Nat. Comms. 7, 12069 (2016) to address the referee's question.

Q3. Fix typo "mater" in abstract.
A3. We have corrected the typo pointed out.

Q4. Fix the first sentence of the second paragraph in Sec. 1 "The effect ...", as it's grammar appears to be broken.
A4. We have identified the missing character “f” and fixed the sentence.

Q5. The factor $\eta$ should be defined in the caption of Fig. 6 for easier readability.
A5. We have added the statement defining $\eta$ as suggested by the referee.

Q6. In the caption of Fig. 7 the difference between lines (2) and (3) is attributed to collision loss during TOF. Is this the only interpretation? Maybe a short discussion is warranted.
A6. We add a short discussion in the text for clarification.

Q7. In the appendix A.3 on GPE simulations I would like to see, for completeness, more details that make it easier for others to reproduce the results:
Q7a. The formula used for $g_{1D}$ should be given.
A7a. We include the expression for $g_{1D}$ as suggested.
Q7b. The form of the external potential used (and relation to $E_r$) should be given (or a relevant reference cited).
A7. We include the expression for the lattice potential and the definition of $E_r$ in the revised manuscript as suggested.
Q7c. The chemical potential $\mu$ shown in Fig. 11 should be defined in terms of the quantities of Eq. (1) (or should it appear in the equation?).
A7c. Fig. 11 shows the results from homogeneous 1D simulations. The chemical potential is directly set by the line density of particles and the 1D interaction parameter. We clarify the chemical potential in this case in the revised manuscript.
Q7d. Parameters of the simulation (how many lattice sites per computational box, discretisation parameters of the GPE) should be added.
A7d. We specify the settings for the GPE simulations in Appendix A.3 in the revised manuscript as suggested.

---

## Round 2 · Referee Report · Anonymous (Referee 1) · 2022-4-6

Report

I am happy with the answers provided to my questions and with the revised manuscript. This work pushes the widely used technique of diffracting matter waves from a standing light wave into the regime of a strongly interacting quantum gas. It raises some interesting question and thus I have no doubt that this work will inspire interesting and valuable follow-up work. I support publication of the current version of the manuscript in SciPost Physics.

---

## Round 2 · Referee Report · Anonymous (Referee 2) · 2022-4-6

Report

The authors addressed all comments and suggestions satisfactorily, including an added discussion of effectively reduced lattice depth, a new version of Fig. 4, updated figures, and corrected typos.

I can recommend the manuscript for publication as-is.

---

## Round 2 · Author Response

We thank the referees for the constructive suggestions. Their comments are addressed with the changes listed below. And more details are described in our replies to the referees.

---

## Round 2 · List of Changes

- According to Referee 1's comment 1 and 5, in Sec. 4.1 Para. 6, we add additional discussion on effective reduction of the lattice depth.
- According to Referee 1's comment 2, in Appendix. A1 Para 6, we state the detail of lattice depth calibration.
- According to Referee 1's comment 3, in caption of Fig.3 and 4, we clarify the calculation of theory curves and the use of free parameters.
- According to Referee 1's comments, in Sec. 4.1 Para 2, explanation given for different lattice depth chosen.
- According to Referee 1's comments, in Fig.4, a zoom on the relevant region is added to the figure, and the caption changed accordingly.
- According to Referee 2's comments, in Sec. 1 Para 2, we add discussion of strongly interacting molecular system.
- According to Referee 2's comment 1, in abstract, we clarify the discussion on mean field simulation.
- According to Referee 2's comment 2, in Sec. 4.2 Para. 5, we comment on secondary collisions.
- According to Referee 2's comment 5, in Fig.6 caption, the factor $\eta$ is explained.
- According to Referee 2's comment 6, in Sec. 4.2 Para 3, we add a discussion on different contributions of loss.
- According to Referee 2's comment 7, in Appendix. A3, we include quantitative information for the GPE simulation.
- Correction of typos.
- Figure improvement according to Referee 1’s comments.

---

## Editorial Decision

published